# Innate Cytokine Induced Early Release of IFNγ and CC Chemokines from Hypoxic Human NK Cells Is Independent of Glucose

**DOI:** 10.3390/cells9030734

**Published:** 2020-03-17

**Authors:** Sonia Y. Velásquez, Bianca S. Himmelhan, Nina Kassner, Anna Coulibaly, Jutta Schulte, Kathrin Brohm, Holger A. Lindner

**Affiliations:** Department of Anesthesiology and Surgical Intensive Care Medicine, Institute for Innate Immunoscience (MI3), University Medical Center Mannheim, Medical Faculty Mannheim, Heidelberg University, 68167 Mannheim, Germany; Sonia.Velasquez@medma.uni-heidelberg.de (S.Y.V.); biancahimmelhan@msn.com (B.S.H.); ninakassner@yahoo.de (N.K.); Anna.Coulibaly@medma.uni-heidelberg.de (A.C.); Jutta.Schulte@medma.uni-heidelberg.de (J.S.); Kathrin.Brohm@merckgroup.com (K.B.)

**Keywords:** glycolysis, hypoxia, inflammation, innate immunity, interferon γ, interleukin 15, natural killer cells, priming

## Abstract

Natural killer (NK) cells are among the first innate immune cells to arrive at sites of tissue inflammation and regulate the immune response to infection and tumors by the release of cytokines including interferon (IFN)γ. In vitro exposure to the innate cytokines interleukin 15 (IL-15) and IL-12/IL-18 enhances NK cell IFNγ production which, beyond 16 h of culture, was shown to depend on metabolic switching to glycolysis. NK effector responses are, however, rapid by comparison. Therefore, we sought to evaluate the importance of glycolysis for shorter-term IFNγ production, considering glucose deprivation and hypoxia as adverse tissue inflammation associated conditions. Treatments with IL-15 for 6 and 16 h were equally effective in priming early IFNγ production in human NK cells in response to secondary IL-12/IL-18 stimulation. Short-term priming was not associated with glycolytic switching but induced the release of IFNγ and, additionally, CCL3, CCL4 and CCL5 from both normoxic and hypoxic NK cells in an equally efficient and, unexpectedly, glucose independent manner. We conclude that release of IFNγ and CC chemokines in the early innate immune response is a metabolically autonomous NK effector program.

## 1. Introduction

Natural killer (NK) cells mediate innate host defense against infection and tumors [1,2,3]. They develop predominantly from the common lymphoid progenitor [4]. NK cell development, survival and homeostasis depend on interleukin 15 (IL-15) [5,6] which also promotes NK cell migration [7,8] and enhances their effector functions in response to secondary stimulation, referred to as priming [9,10,11,12].

Activated NK cells secrete various cytokines [13,14,15], above all interferon γ (IFNγ) which supports antigen presentation [16,17], the CC chemokines CCL3, CCL4 and CCL5 that attract activated T-lymphocytes, monocytes, dendritic cells, eosinophils and more NK cells [18,19] and the major inflammatory cytokine tumor necrosis factor (TNF)α [20]. During infectious inflammation, antigen presenting cells produce the innate cytokines IL-12 [21], IL-15 [22,23] and IL-18 [24] which promote IFNγ production in NK cells [25]. In vitro exposure of human and mouse NK cells to IL-15 in combination with IL-12 and IL-18 for 16 h potentiates their effector functions, including IFNγ production and induces a memory-like NK anti-tumor phenotype [26].

Targeted NK cell migration to infected tissue sites and developing tumors was mostly examined in experimental mice following days to weeks after the onset of the tissue injury or the injection of labeled NK cells [27,28]. However, numbers of NK cells in peritoneal fluid were increased as soon as 12 h after intraperitoneal mouse hepatitis virus infection [29] and labeled NK cells were shown to localize to experimental lung metastases in mice already 16 h after their adoptive transfer [30]. Sherwood and colleagues demonstrated in a murine model of polymicrobial peritonitis by cecal ligation and puncture (CLP) that migration of NK cells from secondary lymphoid organs to the peritoneal cavity occurs in as little as 4–6 h [31,32]. In this model, IL-15 dependent release of IFNγ from NK cells contributes to the pathogenesis of septic shock [33]. Therapeutic administration of IL-15 can, however, also improve survival in CLP mice by reversing innate and adaptive immune dysfunction [34], and IL-15-based agonists are undergoing clinical trials as antitumor and antiviral agents [35].

In the acute phase of inflammation, metabolic demands of immune cells at the site of injury increase. Infiltrating neutrophils consume glucose and oxygen for the production of reactive oxygen species [36]. This and reduced oxygen delivery due to thrombosis and interstitial hypertension promote the development of tissue hypoxia [37,38]. Under these conditions, inflammatory macrophages, T cells and dendritic cells switch to a Warburg-type metabolism reducing pyruvate to lactate to meet anabolic and energy demands [39,40].

Human NK cells assume a glycolytic phenotype during acquisition of functional competence and self-tolerance, a process called licensing or education [41,42]. Although human and mouse NK cells still display low levels of glycolysis and mitochondrial respiration in the resting state, in vitro treatment with IL-15 or combinations of IL-2, IL-12, IL-15 and IL-18 for overnight or longer causes glycolytic switching concomitant with an upregulation of oxidative phosphorylation (OxPhos) and increased reliance of IFNγ production on glycolysis [43,44]. Finlay and coworkers have shown that a proportion of the pyruvate generated through glycolysis in mouse NK cells treated for 20 h with IL-2/IL-12 is converted into mitochondrial citrate that enters the citrate-malate shuttle (CMS) [45]. They concluded that the CMS sustains glycolysis and OxPhos by supplying the necessary cytosolic NAD^+^ and mitochondrial NADH, respectively, and, thereby, supports IFNγ production and cell killing activity [46]. Recent work in mice has shown that NK anti-viral and anti-tumor activity relies on glycolysis in vivo [47,48].

The time required for cytokine induced glycolytic switching to occur (>16 h) [43,44] is relatively long compared to the rapidity of NK cell target site migration [29,30,31,32] and the promptness of their effector responses [2]. NK cells thus reach their target tissues and participate as bona fide innate immune responders in the first line of defense well before cytokine induced glycolytic switching becomes effective. In this early phase, NK cells already need to function in prevalently hypoxic environments in that other cells compete for glucose [36,39,40]. In this regard, it is notable that IFNγ production in mouse NK cells treated for 6 h with IL-12/IL-18 was shown previously to remain unaffected by glucose deprivation or chemical ATP-synthase inhibition by oligomycin [49]. Accordingly, human NK cells primed with IL-15 for 6 h show only a very moderate increase in glycolysis [7]. Nevertheless, such short-term IL-15 exposure and hypoxia transcriptionally upregulate hypoxia inducible factor (HIF-1α) and glycolysis pathways in a synergistic fashion [7]. In agreement with this, treatment of human NK cells with IL-15 for 4 h under chemical hypoxia augments protein levels of HIF-1α [50], the major mediator of cellular adaption to hypoxia [51]. The impact of hypoxia on NK cell glycolysis measured at low oxygen has, however, not yet been reported.

To shed further light on the metabolic requirements of human NK cells during early cytokine exposure, we investigated here how short-term IL-15 priming followed by short-term secondary stimulation with IL-12/IL-18 affects glycolysis and to what extent the cytokine response to this treatment relies on oxygen and glucose availability. Unexpectedly, primed NK cells mounted similarly efficient IFNγ and CC chemokine responses to secondary stimulation under hypoxia, glucose deprivation and both these conditions combined. Priming, secondary stimulation and hypoxia led to only minor increases in glycolysis. Our data suggests that the early release of IFNγ and CC chemokine is a metabolically autonomous NK effector program, well adapted to proceed despite limited oxygen and glucose availability that is characteristic of inflamed tissue sites [37,38].

## 2. Materials and Methods

### 2.1. Cell Isolation and Culture

Approval for this study was obtained by the Ethics Committee Medical Faculty Mannheim (approval number 2016-521N-MA). For all experiments, NK cells were isolated, their purity was evaluated and cells were cultured as described [7]. In brief, immunomagnetic cell separation (NK-Cell Isolation Kit, Miltenyi Biotec, Bergisch Gladbach, Germany) yielded >99% CD45^+^ leukocytes that stained ≥93% for NK cells (CD56^+^CD3^−^) and ≤1% each for T cells (CD3^+^), B cells (CD19^+^), granulocytes (CD15^+^) and monocytes (CD14^+^) by flow cytometry. Peripheral blood of healthy volunteers was used for extracellular flux analysis and buffy coats were used for all other experiments. Isolated cells were maintained at 10^6^/mL in complete RPMI-1640 medium containing 10 mM glucose (Sigma-Aldrich, Munich, Germany) and supplemented with 10% fetal bovine serum (FBS) and 2 mM l-glutamine (Thermo Fisher Scientific, Paisley, UK) in a standard tissue culture incubator (normoxia) or, for experiments at 1% oxygen (hypoxia), in an oxygen-controlled Eppendorf New Brunswick Galaxy 48R CO_2_ incubator (Hamburg, Germany). Alternative to low oxygen, chemical hypoxia was established in some experiments as described [50] using the pan-prolyl hydroxylase inhibitor dimethyloxalylglycine (DMOG) (Selleck Chemicals, Houston, TX, USA) at 50 µM or 1-(5-Chloro-6-(trifluoromethoxy)-1*H*-benzoimidazol-2-yl)-1*H*-pyrazole-4-carboxylic acid (JNJ) (Calbiochem, San Diego, CA, USA) at 20 µM. At these concentrations, DMOG and JNJ were equally efficient in increasing glycolytic rates in primed human NK cells (Appendix A). Experiments with chemical hypoxia included dimethyl sulfoxide vehicle in the normoxia controls.

### 2.2. Cell Treatment and Co-Culture

Recombinant human cytokines were obtained from PeproTech (Rocky Hill, NJ, USA) (IL-4, IL-10, IL-12, IL-13, IL-15), MBL International (Woburn, MA, USA) (IL-18) and R&D Systems (Bio-Techne, Wiesbaden, Germany) (TGF-β). Cytokines were reconstituted in PBS plus 0.1% BSA (Miltenyi Biotec), with the exception of TGF-β which was dissolved in 4 mM HCl plus 0.1% BSA and were stored at −20 °C until use.

For priming experiments, freshly isolated NK cells were left untreated for 16 h before IL-15 (45 ng/mL) was added for another 6 h and cells were collected for further analyses 22 h after seeding. Figure 1 schematizes one extension to this IL-15 priming protocol (Scheme 1) and one modification (Scheme 2), both incorporating a secondary stimulation. Secondary stimulation in Scheme 1 was done with IL-12 (10 ng/mL) and IL-18 (50 ng/mL) alone or in combination (IL-12/IL-18) or with IL-4 (20 ng/mL), IL-10 (10 ng/mL), IL-13 (2 ng/mL) or TGF-β (5 ng/mL). Scheme 2 was exclusively used with IL-12/IL-18. The same respective cytokine concentrations were used throughout unless indicated otherwise.

Scheme 1: IL-15 was added to the culture at 16 h. Secondary cytokines were added at the end of a 6-hour IL-15 priming period for another 4 h in the continued presence of IL-15. Cells were collected for further analyses 26 h after seeding.

Scheme 2: IL-15 was added already at the beginning of the culture and secondary cytokines were added after 16 h for another 6 h. Cells were collected 22 h after seeding and analyzed for IFNγ production by flow cytometry.

In both treatment schemes, cells were kept under normoxia, hypoxia or chemical hypoxia, as indicated, for the complete culture period unless specified otherwise (in Figure 9). Hypoxic cultures were intermittently exposed to ambient oxygen for adding IL-15 and secondary cytokines.

For glucose deprivation experiments, we used supplemented RPMI-1640 without glucose (Sigma-Aldrich). It has to be noted, that supplementation with 10% FBS still introduces low amounts of glucose into the glucose-free medium amounting to approximately 0.67 mM [52]. In the description of our results (Figures 6, 7 and 9 and Appendix A) we do not indicate this small contribution. We refer to 100% glucose-free but supplemented medium as glucose-deficient.

For flow cytometric analysis of intracellular cytokines, brefeldin A (BD Biosciences, San Jose, CA, USA) was added during the final 4 h of the culture experiment according to the manufacturer’s protocol.

### 2.3. Extracellular Flux Assay

Extracellular acidification rate (ECAR) and oxygen consumption rate (OCR) were measured on a Seahorse XFp analyzer (Agilent Technologies, Santa Clara, CA, USA) as described [7]. Mitochondrial carbon source utilization was characterized with the Seahorse XF Mito Fuel Flex Test kit (Agilent Technologies). Parameters of glycolysis and mitochondrial function, respectively, were assessed using the Seahorse XF Glycolysis and Cell Mito Stress Test kits (Agilent Technologies). Additionally, we assessed the velocity of glycolytic flux decay upon glucose withdrawal by monitoring ECAR. Cells pretreated as indicated were washed twice in XF RPMI base medium pH 7.4 supplemented with glutamine but, optionally, glucose (Agilent Technologies) and were immediately subjected to parallel flux measurement with and without glucose.

Flux assays under normoxia were performed in ambient atmosphere. For measurements at 1% oxygen (hypoxia), the XFp analyzer was placed under an oxygen- and temperature-controlled acrylic glass hood constructed by the electrical and mechanical engineering workshop Neuenheimer Feld (Heidelberg University). The hood was equipped with a nitrogen gas line, an MF420-O-Zr oxygen sensor and a GWZ-S2.1 limit signaling device (LogiDataTech, Baden-Baden, Germany).

Generally, duplicate and triplicate determinations were carried out when three and two conditions, respectively, were assayed for comparison in an experiment. Data was analyzed using Seahorse Wave software (Agilent Technologies). During flux measurements, cells were in the presence of the same concentrations of cytokines and/or DMOG or JNJ as during the respective preceding cell treatment, unless conditions were manipulated as indicated.

### 2.4. Flow Cytometry Analysis

Fluorochrome-conjugated monoclonal antibodies (mAbs) were purchased from BD Biosciences (mouse anti-human CD3-FITC [UCHT1], CD56-PE [N901], CD45-PerCP [J.33], IFNγ-FITC or IFNγ-APC [B27] and mouse IgG1κ-FITC [MG1-45] or IgG1κ-APC [MOPC-21] as isotype controls, perforin-PerCP-Cy5.5 and granzyme B-BV510 [GB11]) and from PeloBiotech (Planegg, Germany) (Glut1.RBD (GFP)). For the assessment of GLUT1 surface expression, NK cells were stained with Glut-1.RBD for 20 min at 37 °C followed by staining with cell surface markers for 15 min at RT. For intracellular staining, NK cells were first stained with cell surface markers and sequentially fixed and permeabilized using the BD Cytofix/Cytoperm Fixation/Permeabilization Kit (BD Biosciences) following the manufacturer’s recommendations.

A total of 10^5^ isolated stained NK cells were acquired on a FACSCanto II cytometer using FACSDiva software (BD Biosciences) and analyzed using FlowJo software (Tree Star, Version 10.4.1, Ashland, OR, USA). We gated on singlets in the forward scatter area versus height plot followed by gating on lymphocytes in the sideward scatter area versus forward scatter area plot. Next, we applied sequential gates on biparametric dot plots. For the quantitation of intracellular IFNγ alone, the gating strategy as illustrated in Appendix A included a step for the exclusion of CD3^+^ cells. As these represented less than 1% of all CD56^+^ cells, CD3 staining was omitted in the subsequent experiments for the quantitation of multiple NK markers (Appendix A).We determined marker percentages and median fluorescence intensity (MeFI) values in the CD56^+^ population defined as NK cells.

### 2.5. Western Blotting

Western blotting was performed as recently described [50] with the following antibodies—anti-TPI (ab96696), anti-PGK1 (ab38007) and anti-PDK1 (EPR19573, ab207450) from Abcam (Cambridge, UK) and β-Actin (8H10D10) from Cell Signaling Technology (Danvers, MA, USA).

### 2.6. Bead Array Analysis

We used MILLIPLEX map kits (Merck Millipore, Burlington, MA, USA) on a MAGPIX system (Luminex, Austin, TX, USA) to determine the concentrations of secreted proteins in cell culture supernatants stored at −80 °C. The HCYTOMAG-60K kit was used for IFNγ, TNFα, CCL3, CCL4, CCL5 and VEGF and the HCD8MAG-15K kit for perforin and granzyme B. Duplicate determinations were averaged using MILLIPLEX Analyst software (Merck Millipore).

### 2.7. Statistical Analysis

Data is presented as scatter plots and/or mean values with standard error of the mean, SEM. We used GraphPad Prism software V7.04 (San Diego, CA, USA) for statistical analyses. The Friedman test with Dunn’s test for post-hoc pairwise comparisons was applied to assess differences within experimental groups and the Wilcoxon signed-rank test to compare corresponding samples between groups as indicated. The *p*-values < 0.05 were considered significant.

## 3. Results

### 3.1. Long- and Short-Term IL-15 Priming Equally Support Cytokine Induced IFNγ Production in NK Cells

Enhancement of glycolysis following overnight or longer exposure of NK cells to pro-inflammatory cytokines has been shown to be a requirement for the cytokine induced IFNγ production [43,44]. We first asked whether treatment of human NK cells with IL-15 for only 6 h improved this important NK effector response, induced by subsequent short-term exposure to IL-12/IL-18, as efficiently as prior overnight treatment with IL-15. To this end, NK cells were treated according to the two schemes shown in Figure 1.

Independent of the treatment scheme, neither IL-15, IL-12, nor IL-18 alone enhanced NK cell IFNγ production as judged by the proportions of cytokine positive cells (IFNγ^+^) (Figure 2A,B) and intracellular IFNγ abundance (MeFI) (Figure 2A,C). The three combinations of two cytokines each appeared to, on average, expand the population of IFNγ^+^ cells from below 5% to around 20%. However, only all three cytokines together significantly increased both measures of NK cell IFNγ production in both treatment schemes. Importantly, there was merely a small 1.2-fold increase in the mean proportion of IFNγ^+^ cells through IL-15 pre-treatment for 16 h instead of only 6 h followed by addition of IL-12/IL-18 for 6 h and 4 h, respectively.

### 3.2. Pyruvat Does Not Fuel Respiration in IL-15 Primed and IL-12/IL-18 Stimulated NK Cells

While glycolysis and OxPhos both increase following overnight and longer treatment of NK cells with inflammatory cytokines, short-term cytokine stimulation has little if any metabolic effect [7,43,46]. Nevertheless, priming of human NK cells with IL-15 for just 6 h supported early IFNγ production in response to short-term secondary IL-12/IL-18 stimulation as efficiently as IL-15 pre-treatment for 16 h (Figure 2). Therefore, we next sought to identify the carbon source that fuels mitochondrial respiration in short-term cytokine stimulated human NK cells. Specifically, we considered the use of the glycolytic product pyruvate, of fatty acids and of glutamine upon IL-15 priming for 6 h. To this end, OCR values were monitored and the metabolic pathways that funnel the three fuels into the TCA cycle were sequentially blocked by adding mitochondrial pyruvate carrier (MPC) inhibitor UK5099, glutaminase (GLS) inhibitor BPTES (bis-2-(5-phenylacetamido-1,3,4-thiadiazol-2-yl)ethyl sulfide) and carnitine palmitoyltransferase 1A (CPT1A) inhibitor etomoxir, provided with the Seahorse XF Mito Fuel Flex Test kit. In addition to normoxia, cells were cultured and measurements were done in the presence of DMOG and JNJ to induce the HIF-1α dependent hypoxia response which includes switching from oxidative to glycolytic metabolism [53]. Indeed, chemical hypoxia reduced OCR values throughout (Figure 3) with DMOG showing a more dramatic effect than JNJ (Figure 3A). But temporal profiles appeared otherwise very similar to normoxia suggesting no change in fuel selection through the hypoxia response upon short-term priming with IL-15.

The absence of an effect of MCP inhibition on OCR values (Figure 3A,B) indicates that pyruvate did not represent a significant mitochondrial fuel upon NK cell priming by IL-15. Only when MPC inhibition was followed by inhibition of GLS and CPT1A at the same time, i.e., inhibition of all three pathways collectively, a small drop in OCR values was observed that was significant under normoxia and with DMOG (Figure 3A). Additional Fuel Flex tests in the presence of DMOG and with stepwise inhibition of MPC and GLS (Figure 3B) and of CPT1A and GLS (Figure 3C) did also not identify any significant dependency of respiration on glutamine (GLS) and fatty acids (CPT1A) individually.

In another Fuel Flex experiment, subsequent MPC and GLS/CPT1A inhibition was tested in NK cells cultured under normoxia or with DMOG, primed with IL-15 and additionally exposed for another 4 h to IL-12/IL-18, which corresponds to treatment scheme 1 (Figure 1). In this case, triple-pathway inhibition caused a drop in OCR values that was significant only with DMOG plus secondary cytokine stimulation (Figure 3D).

This data suggests that neither pyruvate, glutamine, nor fatty acids make important contributions to mitochondrial oxygen consumption in IL-15 primed and IL-12/IL-18 stimulated human NK cells individually. A minor reduction of oxygen consumption by collective inhibition of their conversion into mitochondrial fuels, however, indicates that all three pathways can in principle be active.

### 3.3. Glycolytic Protein Expression in Primed NK Cells under Hypoxia

Aside from its role in fueling mitochondrial respiration, glycolysis also supports anabolic reactions by yielding amino acid precursors and through the pentose phosphate shunt which yields NADPH. To thus further probe the significance of glycolysis in human NK cells, we next assessed changes in cellular levels of selected glycolytic proteins in short-term cytokine stimulated human NK cells. We previously validated IL-15/hypoxia induced synergistic upregulation of glycolytic genes by RT-PCR [7]. These included the genes encoding triose-phosphate isomerase (TPI), phosphoglycerate kinase 1 (PGK1) and pyruvate dehydrogenase kinase isozyme 1 (PDK1). Protein level expression for these enzymes, however, was very stable (Figure 4A). Compared to normoxia, IL-15 treatment for 10 h in the presence of DMOG had no effect. Likewise, a 4-hour exposure to IL-12/IL-18 with and without prior IL-15 priming, corresponding to scheme 1 in Figure 1, yielded the same protein levels throughout.

Contrary to the stable abundance of glycolytic enzymes, the proportion of NK cells positive for surface expression of glucose transporter 1 (GLUT1) more than doubled through priming and hypoxic culture (Figure 4B). Extending the 6-hour IL-15 priming period to 10 h further increased GLUT1-positive proportions to around 80% both with and without exposure to IL-12/IL-18 during the 4 h extension. In the presence of IL-15, hypoxia increased mean cell surface levels of GLUT1, reflected by MeFI values, by around 30% (Figure 4C).

### 3.4. Moderate Early Increase in NK Cell Glycolysis through Hypoxia and Inflammatory Cytokines

The observation that neither IL-15 priming for 6 h, nor secondary stimulation with IL-12/IL-18 for 4 h, nor chemical hypoxia changed protein levels of selected glycolytic enzymes in human NK cells (Figure 4A) may agree with the very small effects on glycolysis by 6 h IL-15 priming and hypoxia detected previously during flux measurements performed in ambient atmosphere [7]. However, GLUT1 cell surface expression was increased through priming and hypoxia (Figure 4B,C). This led us to re-evaluate changes in glycolytic parameters in response to priming and secondary stimulation (according to Scheme 1 in Figure 1) under chemical hypoxia and using a hypoxic chamber for flux measurements at 1% O_2_. In the following, glycolytic rate and capacity, respectively, refer to the flux determined before and after adding oligomycin which inhibits ATP synthase, i.e., mitochondrial ATP production.

Overall, ECAR values remained moderate under all conditions tested. Values for capacity were more variable than for rate. Priming of resting NK cells for 6 h with IL-15 led to very low but still noticeable elevations of basal glycolysis and also respiration (Appendix A). Values for capacity were generally more variable than for rate. Figure 5A shows a slight increase in glycolytic rate and capacity through hypoxic versus normoxic IL-15 priming for 6 h. IL-12/IL-18 for 4 h under normoxia led to higher rate and capacity than IL-15 for 10 h (Figure 5B). The combination of IL-15 priming with secondary IL-12/IL-18 stimulation further increased rate. This confirms small but positive early effects on glycolysis for all three cytokines and hypoxia. IL-15, however, also appeared to prevent further enhancement of capacity through secondary IL-12/IL-18 stimulation under normoxia (Figure 5B) but not under hypoxia (Figure 5C). Chemical hypoxia by DMOG and JNJ also moderately enhanced glycolytic rate in IL-15 primed NK cells (Appendix A) and upon further addition of IL-12/IL-18 (Appendix A).

We additionally used the anti-inflammatory cytokines IL-4, IL-10, IL-13 or TGF-β for secondary stimulation of IL-15 primed NK cells but found no effect on glycolytic parameters (Figure 5D–G). These cytokines did not abrogate the positive effect of DMOG on rate, while capacity responses were inconsistent between donors, i.e., between individual experiments.

### 3.5. Early Cytokine Induced IFNγ and Chemokine Responses in NK Cells do not Depend on Glucose

In the innate phase of an immune reaction, NK cells can reach inflamed target tissues in only 4–16 h [29,30,31,32], where they are exposed to myeloid cell derived innate cytokines and possibly experience reduced concentrations of glucose [36,39,40]. Keppel et al. (2015) previously determined equal proportions of IFNγ^+^ mouse NK cells in the presence and absence of glucose following a 6-hour treatment with IL-12/IL-18 [49]. This already suggests relative independence of early IFNγ production in response to these cytokines from glycolysis. Before assessing the functional importance of glycolysis in IL-15 primed human NK cells, we were interested in the velocity at which glycolytic flux, judged by ECAR, declines upon secondary stimulation with IL-12/IL-18, according to scheme 1 in Figure 1 and concomitant deprivation of environmental glucose. Figure 6 shows basal glycolytic rates over time measured immediately after the transfer of normoxic (D1–D3) and hypoxic (D4–D6) NK cells, primed for 6 h with IL-15, into glucose deficient medium containing IL-15 plus IL-12/IL-18. Highly similar responses were observed for both conditions. With glucose, rates increased over approximately the first 45 min (except with D3) before stabilizing around the initial (D1, D2, D4, D5), the elevated (D6) or a slightly reduced value (D3). Without glucose, the first recorded rate was already at 70% or less of the corresponding simultaneous rate with glucose. It further declined exponentially for 60 min, dropping down to and below the 2-deoxyglucose background value for the glucose control, before stabilizing (D1) or continuing to decline at a constant slope (D2–D6).

Notably, the OCR traces corresponding to D1–D3 from the experiment shown in Figure 6 show similar and stable respiratory activities over time both with and without glucose (Appendix A), which agrees with no important role of pyruvate as a mitochondrial fuel (cf. Figure 3).

Sudden glucose deprivation upon secondary stimulation with IL-12/IL-18 led to a rapid decline of glycolysis in IL-15 primed human NK cells (Figure 6). Therefore, we used glucose deprivation to assess the importance of glycolysis for NK effector molecule responses in primed NK cells to 4 h of secondary stimulation. After 6 h of priming, cells were washed and distributed into 5 wells for one control and 4 secondary stimulations at varying glucose concentrations for 4 h (Figure 7). Glucose deprivation had little if any effect on IFNγ production (Figure 7A,B) and CC chemokine release (Figure 7C–E) as well as on intracellular levels of the cytotoxic granule proteins perforin and granzyme B (Figure 7F,G). Contrary to cytokine production and release, secondary stimulation had no effect on perforin and granzyme B. With secondary stimulation and maximal glucose deficiency perforin levels were, however, reduced by 15%. Declining glucose concentrations were further associated with a moderate but apparently steady increase in GLUT1^+^ cells from 59% at 10 mM to 72% at 0 mM glucose. Very similar results were obtained when NK cells were cultured for 5 days at 100 ng/mL IL-15 instead of 6 h at 45 ng/mL IL-15 followed by an equal 4-hour pulse of IL-12/IL-18 at varying glucose concentrations. However, this very long-term IL-15 exposure alone already increased proportions of IFNγ^+^ cells and levels of secreted CCL3, CCL4 and CCL5 (CCL3/4/5) compared to short-term IL-15 (Appendix A).

### 3.6. Hypoxia Hardly Affects Early Cytokine Induced IFNγ Production and Chemokine Release by NK Cells

As for glucose, availability of oxygen may be limited when NK cells reach inflamed target tissues [37,38]. Hence, we assessed the influence of hypoxic versus normoxic culture of human NK cells on the IL-12/IL-18 induced production of IFNγ and, additionally, the release of CCL3/4/5 after 6 h of IL-15. The experiment was conducted using the same cultures as for the assessment of GLUT1 surface expression (Figure 4B,C) the upregulation of which is a sign of functional cellular adaption to hypoxia. However, hypoxia affected neither the proportion of IFNγ^+^ NK cells (Figure 8A), the median intensity of the IFNγ^+^ fluorescence signal (Figure 8B), nor the levels of secreted CCL3/4/5 (Figure 8C–E). In normoxic and hypoxic NK cells, results for the different cytokine treatments showed the same profile for IFNγ and the three chemokines with small reductions through hypoxia by on average 20% for CCL3 and 21% for CCL4.

Chemical hypoxia by DMOG, as an alternative to 1% oxygen, had detrimental effects on the cytokine response of human NK cell to IL-15 priming and secondary stimulation with IL-12/IL-18 (Appendix A). Consequently, we discontinued the use of DMOG in functional assays of NK cell responses.

### 3.7. Efficient Early Cytokine Release from Hypoxic NK Cells Despite Glucose Deprivation

IFNγ and CC chemokine responses by short-term IL-15 primed and IL-12/IL-18 stimulated human NK cells were independent of glucose (Figure 7) and essentially as efficient under hypoxia as under normoxia (Figure 8). We reasoned that hypoxia, however, reduces mitochondrial energy production and that hypoxic NK cells thus tap into glycolysis to maintain effector protein levels. To assess the actual importance of glucose availability for NK effector protein levels under hypoxia, human NK cells were pre-cultured for 16 h and IL-15 primed for 6 h under normoxia and hypoxia followed by, in both cases, hypoxic secondary stimulation for 4 h with IL-12/IL-18 at varying concentrations of glucose. Effects on NK effector protein responses including IFNγ, CC chemokines, perforin, granzyme B and additionally TNFα and VEGF, as well as on GLUT1, are summarized in Figure 9.

Following secondary hypoxic stimulation of differentially pre-cultured/primed NK cells, we observed around 13% lower mean proportions of IFNγ^+^ NK cells for hypoxic than for normoxic pre-culture/priming regardless of glucose concentration (Figure 9A). Maximal glucose deficiency also reduced mean IFNγ^+^ proportions by approximately 11% which reached statistical significance only for normoxic pre-culture/priming. Corresponding reductions were more pronounced for mean IFNγ MeFI values (Figure 9B). Following secondary stimulation, these were decreased by 18–29% through hypoxic pre-culture/priming and by 26% and 35% through maximal glucose deficiency, respectively, after normoxic and hypoxic pre-culture/priming. IL-12/IL-18 stimulated IFNγ secretion from IL-15 primed NK cells remained, however, equally high and constant with declining glucose and for both normoxic and hypoxic pre-culture/priming (Figure 9C).

Mean levels of secreted CC chemokines induced by secondary IL-12/IL-18 stimulation appeared unchanged by glucose deprivation (Figure 9D–F). Hypoxic versus normoxic pre-culture/priming reduced levels of CCL3 by on average 21% (Figure 9D) and of CCL4 by 12% (Figure 9E), whereas CCL5 levels were not influenced (Figure 9F).

Cellular abundance for both perforin (Figure 9G) and granzyme B (Figure 9H) was stable throughout, i.e., it was unchanged by IL-12/IL-18 stimulation, by oxygen availability during pre-culture/priming and by glucose deprivation.

Mean levels of TNFα secreted from IL-15 priming controls were very low (<10 pg/mL). They were consistently elevated by secondary IL-12/IL-18 stimulation 53-fold for normoxic and 33-fold for hypoxic pre-culture/priming. Independent of glucose concentration, this resulted in on average 38% lower TNFα levels for hypoxic than normoxic pre-culture/priming (Figure 9I).

Mean levels of VEGF secreted from control NK cells were also very low (<7 pg/mL) and were consistently elevated 2.6-fold by secondary IL-12/IL-18 stimulation independent of both pre-culture/priming oxygen levels and secondary stimulation glucose levels (Figure 9J).

As a metabolic marker, proportions of GLUT1^+^ cells in IL-15 controls and additionally IL-12/IL-18 stimulated NK cells were equally high and independent of pre-culture/priming oxygen levels (Figure 9K). They remained stable at this high level throughout when glucose was withdrawn.

## 4. Discussion

A central observation of this study was that priming of peripheral blood human NK cells with IL-15 for only 6 h supported the early IFNγ response to secondary IL-12/IL-18 stimulation as efficiently as a prior 16-hour exposure to IL-15 (Figure 2). Priming and secondary stimulation in such a narrow time frame induced cellular release of IFNγ and CCL3/4/5 in a manner independent from concomitant glucose and oxygen availability (Figure 9C–F). It is unexpected that hypoxic NK cells do not rely on glycolysis as a non-mitochondrial way of producing energy for early cytokine production and release. Glucose and oxygen independence of early cytokine release also contrasts the long-term (>16 h) dependency of NK effector functions on glycolysis and the association of long-term exposure to inflammatory interleukins with enhanced OxPhos as established previously [43,44].

IFNγ production is one of the NK effector functions in response to long-term cytokine stimulation that depends on glycolytic switching [43,44]. By contrast, the metabolic requirements for this important NK effector function in the immediate early phase of an innate immune response are unknown. To address this gap, we limited in vitro cytokine exposure of NK cells to a maximum of 10 h (6 h IL-15 plus 4 h IL-12/IL-18), additionally, considering environmental changes typically associated with the development of tissue inflammation, namely, shortage of glucose [36] and oxygen [37,38].

Besides granulocytes, NK cells are thought to be among the first innate immune cells to translocate from blood into inflamed tissues [54] in as little as 4–16 h [29,30,31,32]. Possibly already en route and eventually upon arrival, they are exposed to elevated levels of inflammatory cytokines. IL-15 is for instance trans-presented to NK cells in complex with the IL-15Rα subunit [55] already by the endothelium, which supports their trans-endothelial migration [56,57,58], and by antigen presenting cells that additionally respond to NK cell contact with elevated release of IL-12 and IL-18 [9,59]. Despite its inherent limitations, in vitro cell culture has proven useful in dissecting the roles of environmental glucose and oxygen in NK cell regulation by cytokines [7,49]. Arguably, the actual degree and duration of exposure to environmental cues including cytokines in vivo depends amongst others on the nature and the location of the causative injury, which we do not attempt to mimic in our in vitro approach. Among the many complex interactions that cells experience in vivo, we model exclusively the influence of IL-12, IL-15 and IL-18, however, in a presumably meaningful order and timeline. We focus on the events prior to NK cell contact with stressed target cells and consider production of IFNγ and CC chemokines by NK cells already in this early phase as highly relevant readout because they support both immediate innate and later adaptive immune responses [16,17,18,19]. Our choice to apply a commonly used high dose of IL-15 (45 ng/mL) for priming is based on only very moderate short-term priming effects through low doses [60,61]. In the following, the terms priming and secondary stimulation, respectively, refer to treatment with IL-15 for 6 h and subsequent addition of IL-12/IL-18 for another 4 h. Because the cumulative treatment time according to this scheme (Scheme 1 in Figure 1) was 10 h, priming controls were also exposed to IL-15 for 10 h.

After demonstrating efficient short-term IFNγ production by primed human NK cells in response to secondary stimulation (Figure 2), we asked whether this effect was associated with glycolysis. We first assessed mitochondrial fuel selection by NK cells. The absence of an effect by the MPC inhibitor UK5099 on oxygen consumption following priming and secondary stimulation, without and with chemical activation of HIF-1α, a major inducer of glycolysis [53], indicates that respiration did not depend on the mitochondrial import of the glycolytic product pyruvate (Figure 3). The observation, that only the collective inhibition of pyruvate, fatty acid and glutamine entry into the TCA cycle slightly reduced oxygen consumption, raises the question what eventually fuels mitochondrial respiration upon priming and secondary stimulation. Interestingly, IL-2/IL-12 stimulation of murine NK cells for 20 h enhances OxPhos and glycolysis, at least partly, through the CMS which bypasses the TCA cycle but still relies on mitochondrial pyruvate [45]. We have previously reported that treatment of human NK cells for 6 h with IL-15, i.e., priming, induces expression of the glycerol-3-phosphate and malate-aspartate shuttle genes [7]. Therefore, we propose that MPC did not affect cellular oxygen consumption because one or both of these two shuttle systems maintained oxygen consumption through the transfer of electrons from cytosolic NADH into the mitochondrial respiratory chain, effectively, bypassing the TCA and not requiring mitochondrial pyruvate. Eventually, our analysis of short-term fuel selection did not reveal a preferential carbon fuel for respiration in NK cells following priming and secondary stimulation. To hence further probe the importance of glycolysis, we next analyzed short-term changes in the expression of glycolytic proteins and glycolytic flux as indicators of metabolic changes induced by cytokines and hypoxia.

We previously reported synergistic upregulation of HIF-1α by IL-15 priming and chemical hypoxia [50] as well as of glycolytic gene transcription by IL-15 priming and hypoxia in human NK cells [7]. Here, these responses did however not translate into concordant changes in cellular protein levels (Figure 4A). These were instead remarkably stable throughout for the selected enzymes TPI, PGK1 and PDK1 after NK cell priming and after secondary stimulation, both under normoxia and chemical hypoxia. One explanation might be that these stimuli regulate mRNA and protein levels on different time scales. IL-15, e.g., was shown to dramatically increase granzyme B production within 6 h from mRNAs pre-existing at high levels in mouse NK cells, whereas it took 48 h to achieve a comparable increase for perforin [10]. One may also speculate that stable TPI, PGK1 and PDK1 protein levels were due to a regulated translational delay through a post-transcriptional block or, alternatively, that increased glycolytic gene transcription is required to maintain stable enzyme levels in a compensatory fashion. The absence of changes in glycolytic enzyme levels, however, also suggests that enhanced cellular GLUT1 positivity (Figure 4B) was rather responsible for the moderate increases in glycolytic flux through priming (Appendix A) and secondary stimulation (Figure 5B,C) and enhanced GLUT1 cell surface abundance (Figure 4C) for moderately increased flux through hypoxia and chemical hypoxia (Figure 5A,D–G, Appendix A). The role of GLUT1 as a major determinant of glycolytic activity in NK cells agrees with a recent study performed in immortalized baby mouse kidney epithelial cells that identified glucose import as one of four key flux-controlling steps in glycolysis [62]. Notably, glycolysis was not affected when NK cells were treated for 6 h with IL-4, IL-10, IL-13 or TGF-β instead of IL-15 (Figure 5D–G) indicating that the ability to support glycolysis in this setting is characteristic to inflammatory cytokines. Overall, the enhancements of glycolysis through priming, secondary stimulation, hypoxia and chemical hypoxia in human NK cells were equally moderate suggesting no interaction between these factors.

Collectively, the observed independence of cellular respiration from pyruvate (Figure 3), the stable levels of glycolytic enzymes (Figure 4A) and the moderate increases in glycolytic flux (Figure 5) in response to priming, to secondary stimulation and to hypoxia, likely mediated through upregulation of GLUT1, did not reveal a correlation between glycolysis and IFNγ production. This does not yet preclude potential functional importance of low levels of glycolysis for the short-term cytokine response in NK cells which we next evaluated by manipulating cell culture concentrations of glucose and oxygen. Before, it needs to be mentioned that, as a limitation, our study does not present a conclusive assessment of the importance of glutaminolysis and fatty acid oxidation for human NK effector functions. This will require determining changes in the quantities of metabolic intermediates, e.g., by mass spectrometry, as well as of the involved metabolic proteins.

Transfer of primed human NK cells into glucose-deficient medium for secondary stimulation rapidly suspended glycolytic flux under both normoxia and hypoxia NK cells (Figure 6). In these experiments, supplementary FBS likely introduced a small amount of glucose. However, we previously determined constant mean glucose concentrations of very close to 10 mM in the supernatants of hypoxic, primed human NK cells and controls [7] which is the original glucose concentration in the standard RPMI-1640 medium used. Therefore, we regard FBS glucose in our experiments neglectable.

IFNγ production induced by secondary stimulation in primed human NK cells was independent from glucose availability (Figure 7A,B). The same was true for the release of CCL3/4/5 (Figure 7C–E) and, largely, also for intracellular levels of perforin and granzyme B (Figure 7F,G). We consider it unlikely that potential glucose stores in primed NK cells fully replaced exogenous glucose. The data rather indicates that glycolysis is not a driver of the short-term cytokine response induced by secondary cytokine stimulation in primed human NK cells. This agrees with no change in IFNγ positivity in mouse NK cells through glucose withdrawal during a 6-hour treatment with IL12/IL-18 [49]. We found that NK cells effector molecule levels also retained independence from glucose availability during secondary stimulation following a 5-day exposure to high levels of IL-15. Here, only perforin levels and, possibly, release of CCL3 were sensitive to glucose deprivation (Appendix A). This is of interest because IL-15 exposure for several days may functionally exhausted NK cells [63].

Cells resort to glycolysis when mitochondrial energy production declines such as at low oxygen. Hypoxia, in principle, thus provides another means of assessing the importance of glycolytic metabolism. As priming itself, hypoxic versus normoxic human NK cells, however, showed only moderately increased glycolysis following priming (Appendix A and Figure 5A) and hypoxia did not interact with secondary stimulation (Figure 5C). The same applied to chemical hypoxia (Appendix A). The moderate increase in glycolysis by hypoxia may still have supported levels of IFNγ production at 1% oxygen that remained comparable to normoxia levels (Figure 8A,B). This data, however, indicates that mitochondrial energy production that obligatorily consumes oxygen was not important for the IFNγ response. This interpretation can be extended to the metabolic requirements for the release of the chemokines CCL5 and, largely, also CCL3 and CCL4 (Figure 8C–E). This data and the lack of additional glycolytic reserve through secondary stimulation in primed compared to non-primed human NK cells also conform to unchanged IFNγ positivity in mouse NK cells treated for 6 h with IL-12/IL-18 in the presence of oligomycin [49]. DMOG, which has gained interest as a pharmacological anti-inflammatory hydroxylase inhibitor [38], however, interfered with IFNγ production and CCL3/4/5 release (Appendix A) highlighting functional differences between low oxygen and chemical hypoxia.

In mouse NK cells, inhibition of either mitochondrial ATP synthase by oligomycin or of glycolysis by 2-deoxyglucose during a 6-hour IL-12/IL-18 treatment, reportedly, reduced cellular ATP concentrations by half [49]. Possibly, glucose deprivation and hypoxia diminished human NK cell energy production in our study to a similar extent without compromising the cytokine response (Figure 7 and Figure 8). We hypothesized that cytokine production and release by NK cells, induced by priming and secondary stimulation, still requires significant energy production which may be compromised at inflamed tissue sites due to concomitant local hypoxia [37,38] and low glucose [36,39,40]. Therefore, we subjected human NK cells pre-cultured and primed under normoxia and hypoxia to hypoxic secondary stimulation at declining concentrations of glucose and compared resultant effector molecule responses (Figure 9). In the following, the terms hypoxia/hypoxic and normoxia/normoxic refer to the differential oxygen concentrations during pre-culture and priming. Measures of IFNγ production were only slightly impaired through simultaneous hypoxia and glucose deprivation (Figure 9A,B). But most strikingly, NK cells consistently released relatively high concentrations of IFNγ regardless of concomitant oxygen and glucose availability (Figure 9C). Likewise, neither for normoxic nor hypoxic NK cells, glucose deprivation affected the release of CCL3/4/5 (Figure 9D–F), TNFα (Figure 9I) and VEGF (Figure 9J) and levels of intracellular perforin and granzyme B (Figure 9G,H). The influence of hypoxia on the levels of secreted IFNγ, TNFα, CCL3, CCL5 and VEGF observed here was consistent with reported results obtained with human NK cells stimulated exclusively with IL-12/IL-18 for 20 h under normoxia and hypoxia [64]. Overall, the simultaneous independence from glucose and oxygen of the early cytokine response and of the maintenance of cytotoxic granule proteins in primed human NK cells in response to secondary stimulation shows that these cellular processes require little if any active energy production.

## 5. Conclusions

Our result that the early release of IFNγ and CCL3/4/5 from short-term IL-15 primed and IL-12/IL-18 stimulated human NK cells is unaffected by simultaneous glucose deficiency and hypoxia supports the concept of metabolic autonomy during regulatory NK cell polarization [65], specifically, during the early innate immune response (Supplemental Discussion). We propose that cytokine induced switching to glycolysis and OxPhos to support cytotoxic activity [43,44] succeeds in a time-dependent manner. Noteworthily, engagement of the activation receptor NK1.1. on mouse NK cells by a plate-bound antibody for 6 h, contrary to IL-12/IL-18 exposure for the same duration, induced IFNγ production in a glucose and OxPhos dependent manner [49]. This indicates that the metabolic requirements of IFNγ production by NK cells are stimulus dependent and may suggest that target cell contact triggers much more rapid metabolic switching than innate cytokines. Nevertheless, a quantitative understanding of NK cell metabolism will be necessary to better judge the importance of cellular energy and intermediary metabolism for certain anabolic and effector activities and whether alternative ways of energy storage may function in NK cells, such as the creatine kinase energy system, which supports production of IL-2 and IFNγ in T lymphocytes [66].

## Figures and Tables

**Figure 1 cells-09-00734-f001:**
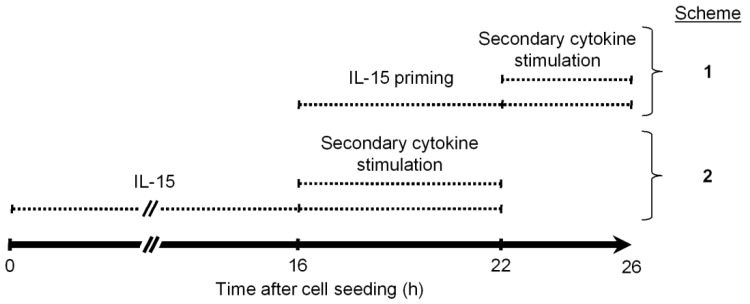
Timing of cytokine treatment. Natural Killer (NK) cells were isolated from human blood, seeded and cytokine treated according to two schemes. Scheme **1** was applied in all experiments and was compared to scheme **2**, exclusively, in an analysis of IFNγ production (Figure 2). Parallel culture manipulations at 16 and 22 h in both schemes simplified the experimental workflow.

**Figure 2 cells-09-00734-f002:**
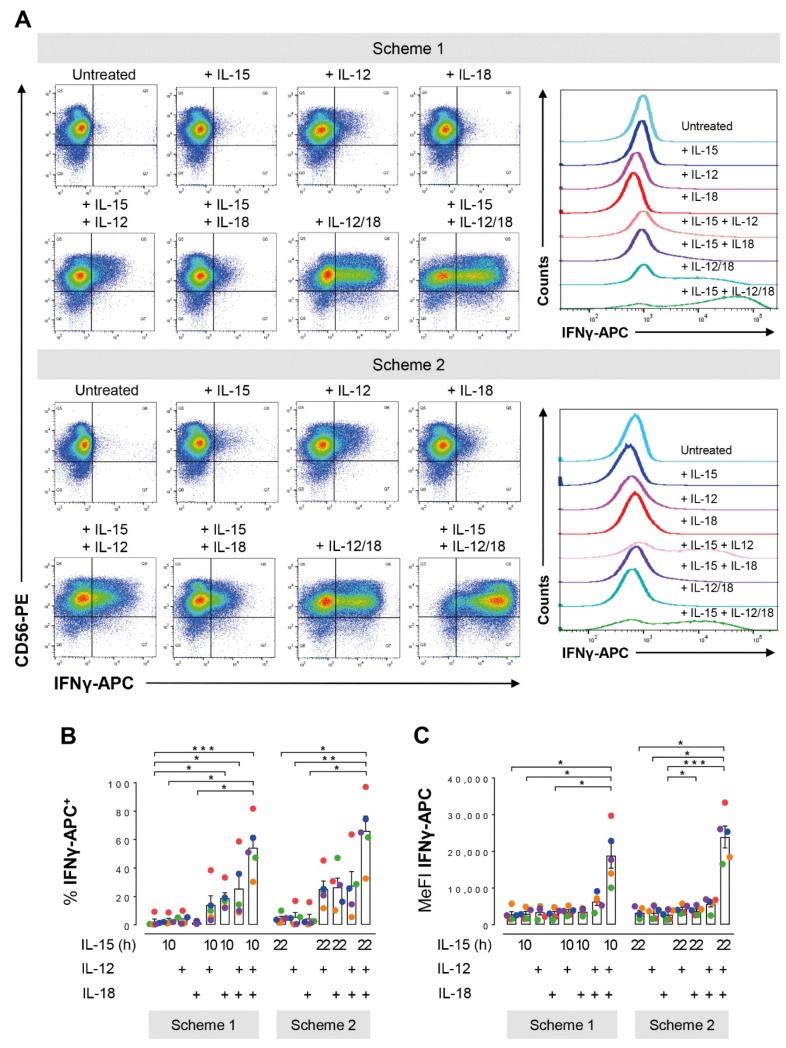
Short- and long-term pre-treatments with IL-15 equally support IFNγ production in IL-12/IL-18 stimulated human NK cells. NK cells from buffy coats were seeded in parallel and treated according to the two schemes shown in Figure 1. Scheme 1: IL-15 was added 16 h after seeding for another 10 h. IL-12 and/or IL-18 were added for the final 4 h of the culture. Scheme 2: Cells were cultured for 22 h with IL-15 present from the beginning and IL-12 and/or IL-18 were added for the final 6 h. (**A**) CD56-PE versus IFNγ-APC dot plots exemplify the distribution of IFNγ producing cells from one representative buffy coat in response to treatments according to Scheme 1 (top) and Scheme 2 (bottom). Stacked histograms on the right display the corresponding IFNγ intensity distributions in the CD56+ population defined as NK cells. Pooled flow cytometric data from 5 independent experiments for (**B**) the proportions of IFNγ^+^ cells and (**C**) MeFIs is shown as mean values ± SEM (bars) and scatter plots in a color scheme to identify data from different experiments. Within a given scheme, statistical significance of mean differences was determined with the Friedman test with Dunn’s test for post-hoc pairwise comparisons. * *p* < 0.05, ** *p* < 0.01, *** *p* < 0.001. Corresponding treatments in Schemes 1 and 2 were compared with the Wilcoxon signed-rank test but none reached the level of statistical significance.

**Figure 3 cells-09-00734-f003:**
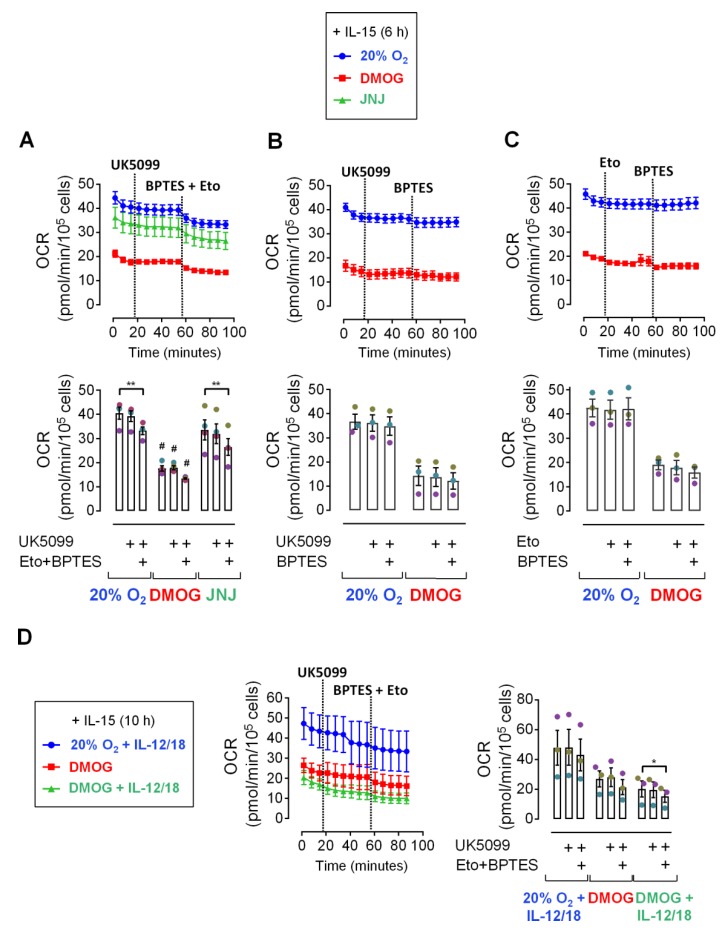
Carbon fuel dependency of oxygen consumption in primed human NK cells. (**A**–**C**) NK cells from 3 or 4 donors were cultured under normoxia (20% O_2_) in the absence or presence of DMOG or JNJ. After 16 h, cells were primed with IL-15 for 6 h (legend on top). Oxygen consumption rate (OCR) values were subsequently acquired over time in the continued presence of IL-15 and with or without chemical hypoxia. The first three measurements were performed under basal conditions followed by the sequential injections of the MPC inhibitor UK5099 (2 µM), the GLS inhibitor BPTES (3 µM) and the CPT1A inhibitor etomoxir (4 µM). In panel (**D**), NK cells were IL-15 primed as in (**A**–**C**) and were cultured for another 4 h in the continued presence of IL-15 and chemical hypoxia and additionally IL-12 and IL-18 (legend to the left). Respective culture conditions were maintained during OCR measurements. The upper parts of panels (**A**–**C**) and the left part of (**D**) show OCR traces based on averaged biological replicates ± SEM with inhibitor injections indicated by dotted lines. The lower (**A**–**C**) or right (**D**) part of these panels displays the last recording before the first injection (baseline) and before the second injection (with first inhibitor) as well as the last recording (with all inhibitors) for the culture conditions indicated below the diagram. Data is shown as mean values ± SEM (bars) and scatter plots in a muted color scheme to identify data from same donors, i.e., independent experiments. Statistical significance of mean differences was determined with the Friedman test with Dunn’s test for post-hoc pairwise comparisons. * *p* < 0.05 and ** *p* < 0.01 for inhibitor effects under same culture conditions, # *p* < 0.05 for comparisons to corresponding experimental time points, i.e., inhibitor compositions, in the normoxia controls which only reached the significance threshold for the DMOG versus normoxia comparison in (**A**).

**Figure 4 cells-09-00734-f004:**
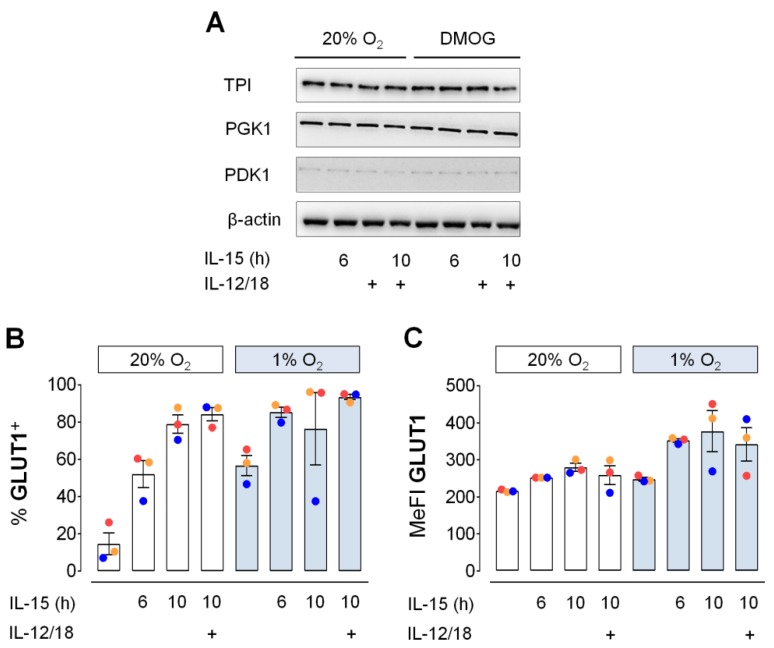
Glycolytic protein expression in primed human NK cells. (**A**) NK cells from buffy coats were cultured under normoxia (20% O_2_) or chemical hypoxia (DMOG). IL-15 was added at 16 h followed by IL-12 and IL-18 (IL-12/18) at 22 h. Cells were collected at 26 h and analyzed by anti-TPI, -PGK1 and -PDK1 immunoblotting using anti-β-actin as loading control. One representative experiment of 3 performed is shown. (**B**,**C**) IL-15 was added to NK cell cultured under normoxia (20% O_2_) or hypoxia (1% O_2_) at 16 h and IL-12/18 at 22 h. Cells were collected at either 22 h (6 h IL-15 only) or at 26 h (untreated controls and 10 h IL-15 with or without 4 h IL-12/18)**.** Flow cytometric data for (**B**) the proportion of GLUT1^+^ NK cells and (**C**) GLUT1 MeFI is represented as mean values ± SEM (bars) and scatter plots with a color scheme to identify data from same buffy coats representing 3 independent experiments. The level of statistical significance was not reached when applying the Friedman test with Dunn’s test for post-hoc pairwise comparisons and the Wilcoxon signed-rank test to compare corresponding cytokine treatments under normoxia and hypoxia.

**Figure 5 cells-09-00734-f005:**
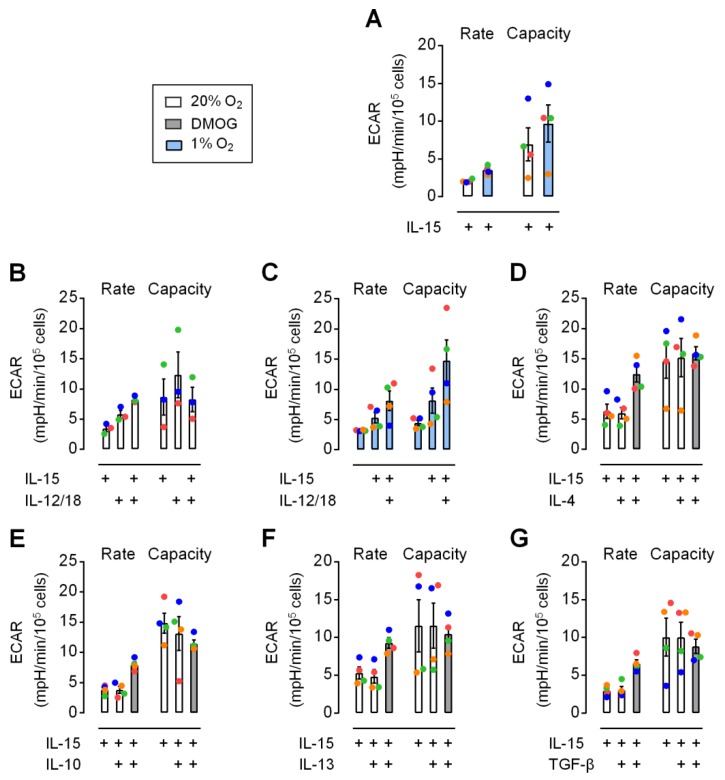
Hypoxia and IL-12/IL-18 moderately increase glycolysis in IL-15 primed human NK cells. NK cells from 3 or 4 donors were cultured for 16 h, IL-15 primed for 6 h and cultured for another 4 h in the additional presence of different cytokines as indicated. During both cytokine treatments and measurements of extracellular acidification rate (ECAR) values, cultures were kept under normoxia (20% O_2_, open bars), hypoxia (1% O_2_, blue bars) or chemical hypoxia (DMOG, gray bars). Panel (**A**) shows the effect of hypoxia and panel (**B**) and (**C**) the effect of IL-12 combined with IL-18 (IL-12/18) under normoxia and hypoxia, respectively. Alternative to IL-12/18, NK cells primed with IL-15 in the absence or presence of DMOG were treated with (**D**) IL-4, (**E**) IL-10, (**F**) IL-13 and (**G**) TGF-β. The level of statistical significance was not reached when applying the Friedman test with Dunn’s test for post-hoc pairwise comparisons.

**Figure 6 cells-09-00734-f006:**
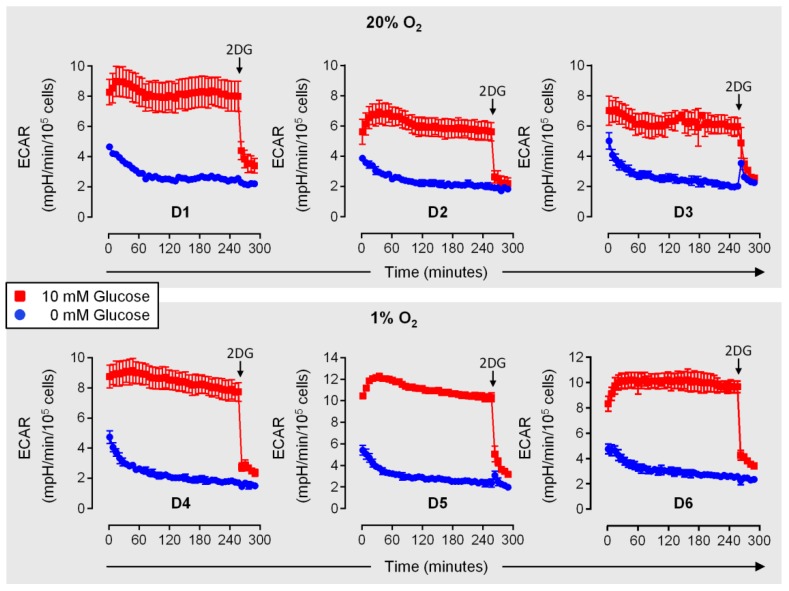
Glycolysis in human NK cells rapidly declines upon glucose deprivation. NK cells were isolated from 6 donors and cultured under normoxia (20% O_2_, D1–D3) or hypoxia (1% O_2_, D4–D6). At 16 h, IL-15 was added for another 6 h. At 22 h cells were transferred into glucose-containing (red) or glucose-deficient (blue) medium in the continued presence of IL-15 plus IL-12/IL-18 and were immediately subjected to extracellular acidification rate (ECAR) measurements. For D1–D3, determinations were done under normoxia and for D4–D6 under hypoxia. Toward the end of the experiment, glycolysis was inhibited by adding 50 mM 2-deoxyglucose (2DG) to all samples (arrow). ECAR traces are based on averaged technical triplicates ± SEM.

**Figure 7 cells-09-00734-f007:**
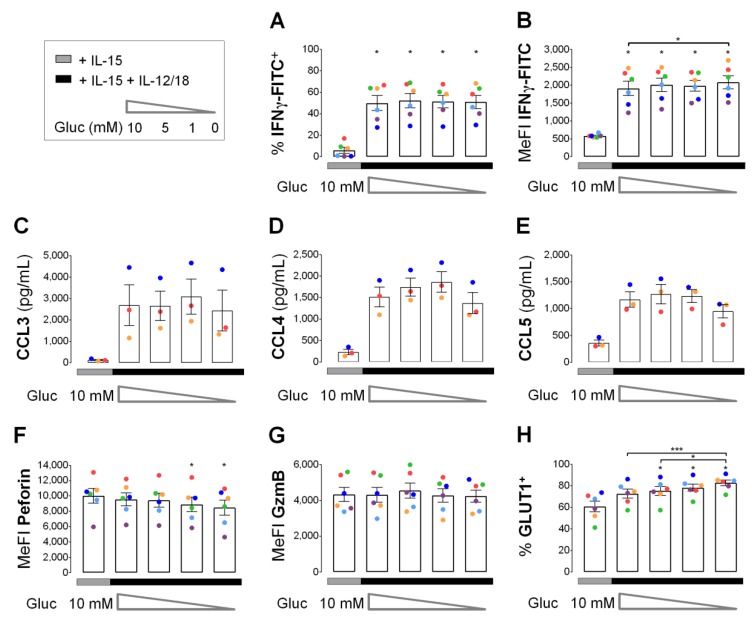
Early cytokine responses in IL-15 primed human NK cells in response to IL-12/IL-18 do not require glucose. NK cells from 3 or 6 buffy coats were cultured for 16 h and primed with IL-15 for another 6 h. Cells were washed with glucose-deficient medium and were re-seeded either in fresh medium with IL-15 and glucose as a control (gray horizontal) or in medium containing IL-15 plus IL-12 and IL-18 (IL-12/18) and glucose concentrations declining from 10 mM to 0 mM (black horizontal). Four hours after re-seeding, cells were collected and analyzed for the proportion of (**A**) IFNγ^+^ cells, (**B**) IFNγ MeFI, secreted (**C**) CCL3, (**D**) CCL4 and (**E**) CCL5, cellular (**F**) perforin and (**G**) granzyme B (GzmB) MeFI and (**H**) GLUT1^+^ cells. Data is shown as mean values ± SEM (bars) and scatter plots in a color scheme to identify data from independent experiments. Within the IL-15 + IL-12/18 group, statistical significance of mean differences was determined with the Friedman test with Dunn’s test for post-hoc pairwise comparisons. Each condition with IL-15 + IL-12/18 was compared individually to the IL-15 control with the Wilcoxon signed-rank test. * *p* < 0.05, ** *p* < 0.01, *** *p* < 0.001.

**Figure 8 cells-09-00734-f008:**
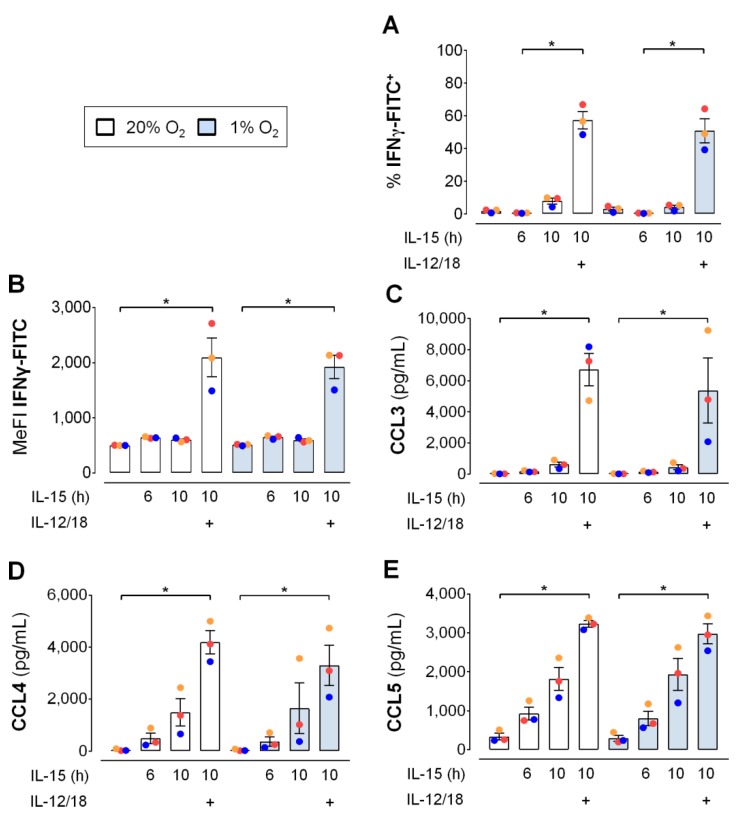
Hypoxia does not affect early IFNγ production and CC chemokine release by IL-15 primed human NK cells in response to IL-12/IL-18. NK cells from 3 buffy coats were seeded and cultured in parallel under normoxia (20% O_2_, open bars) and hypoxia (1% O_2_, blue bars). IL-15 was added at 16 h and IL-12 and IL-18 (IL-12/18) at 22 h. Cells and supernatants were collected at either 22 h (6 h IL-15 only) or at 26 h (untreated controls and 10 h IL-15 with or without 4 h IL-12/18) and analyzed for (**A**,**B**) cellular IFNγ production and levels of secreted (**C**) CCL3, (**D**) CCL4 and (**E**) CCL5. Data is shown as mean values ± SEM (bars) and scatter plots in a color scheme to identify data from independent experiments. Within the normoxia and hypoxia group each, statistical significance of mean differences was determined with the Friedman test with Dunn’s test for post-hoc pairwise comparisons. * *p* < 0.05. Corresponding treatments under normoxia and hypoxia were compared with the Wilcoxon signed-rank test but none reached the level of statistical significance.

**Figure 9 cells-09-00734-f009:**
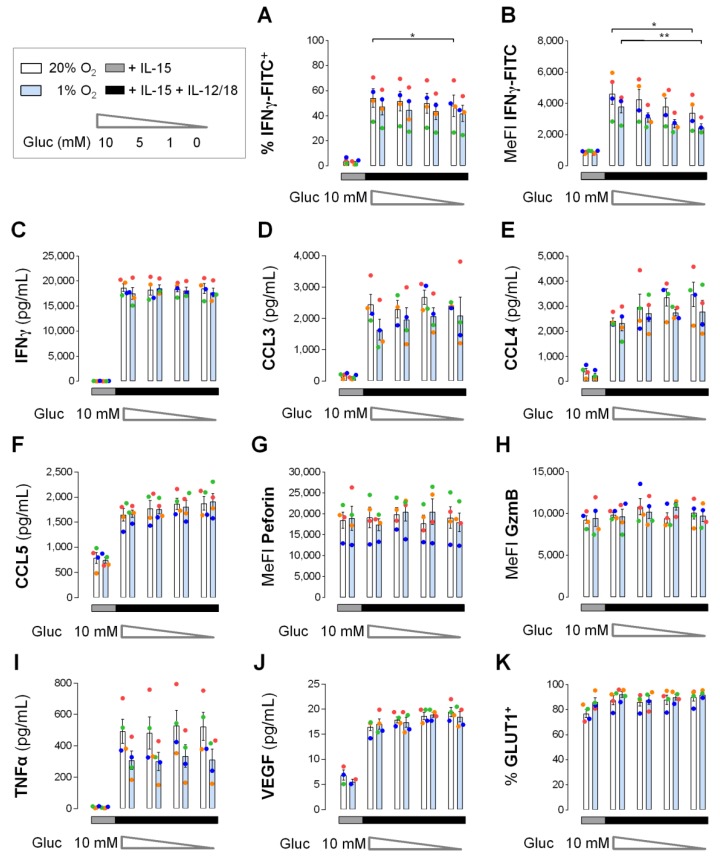
Effector protein levels in human NK cells under concurrent hypoxia and glucose deprivation. NK cells from 4 buffy coats were pre-cultured for 16 h and primed for 6 h with IL-15. For this 22-hour period, one aliquot of cells each was maintained under normoxia (white) and a second aliquot under hypoxia (blue). After priming, both aliquots were washed with glucose-deficient medium and re-seeded into 5 wells each, one well with control medium containing IL-15 and glucose (gray horizontal) and 4 wells with IL-15 plus IL-12/IL-18 and glucose concentrations declining from 10 mM to 0 mM (black horizontal). Controls and secondary stimulations were incubated together for another 4 h under hypoxia. Cells were collected and analyzed for the proportion of (**A**) IFNγ^+^ cells, (**B**) IFNγ MeFI, secreted (**C**) IFNγ, (**D**) CCL3, (**E**) CCL4, (**F**) CCL5, cellular (**G**) perforin and (**H**) granzyme B (GzmB) MeFI, secreted (**I**) TNFα and (**J**) VEGF and (**K**) GLUT1^+^ cells. Data is shown as mean values ± SEM (bars) and scatter plots in a color scheme to identify data from independent experiments. Within the IL-15 + IL-12/18 group, statistical significance of mean differences between glucose concentrations and between normoxic and hypoxic pre-treatment was determined with the Friedman test with Dunn’s test for post-hoc pairwise comparisons. * *p* < 0.05, ** *p* < 0.01. Each condition in the IL-15 + IL-12/18 group was compared individually to the IL-15 control with the same pre-treatment oxygen level with the Wilcoxon signed-rank test but none reached the level of statistical significance.

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
