# Peer review of "Innate Cytokine Induced Early Release of IFNγ and CC Chemokines from Hypoxic Human NK Cells Is Independent of Glucose"

_cells, 2020, doi:10.3390/cells9030734_

Round 1

Reviewer 1 Report

The manuscript by Velasquez et al. investigates the early cytokine response of human NK cells during oxygen- and glucose deprivation. The major findings are:

  • Treatments with IL-15 for 6 and 16 h are both effective in priming early cytokine release in human NK cells in response to secondary stimulation with IL-12/IL-18.

  • Short-term priming induces the release of IFNγ and,additionally, CCL3, CCL4, and CCL5 from both normoxic and hypoxic NK cells in a glucose-independent manner.

  • Short-term priming of human NK cells is not associated with glycolytic switching

The manuscript is well written, the experiments thoroughly executed and the study is certainly of interest, even more so since the role of low oxygen in NK cell metabolism and function in the literature is quite controversial if not confusing. Yet, the manuscript rather adds to this confusion and does not really clarify the actual controversy. Moreover, the manuscript stays rather descriptive and does not provide a lot of mechanistic insight.  The impact of the study is likely to increase by addressing the following issues:

-The above-mentioned confusion is largely due to different experimental setups to study the impact of hypoxia on NK cell function and metabolic changes. The authors use a set of sophisticated but complex but somewhat unique incubation scheme (e.g. different priming schemes followed by secondary activation) that renders the findings hardly comparable to previous findings on the effect of hypoxia on NK cell activation (e.g. Sarkar et al. 2013). Classical NK activation assays with IL-12/IL-18 stimulation subsequent IFNγ measurement run over 4 to 6 hours. The authors could include this experimental into their study. Although, rather simple, this layout is widely used to study NK activation.

-Along these lines, in addition to cytokine release, the actual killing of target cells is a hallmark feature of NK cells. The authors should therefore perform killing assays under their particular incubation scheme. Even more so since the impact of hypoxia on NK cell function/cytotoxicity seems to be most obvious in an actual killing assay upon exposure to target cells (Sarkar et al. 2013). Likewise, in addition to cytokine activation, NK cells are activated or inhibited in a ligand-dependent manner. Ideally, this could be addressed.

The adaptive response to low oxygen depends on the HIF transcription factor family, with HIF1 and HIF2 as the most prominent members. This pathway also seems to play a role in NK cell activation in normoxia and hypoxia (Krzwinska et al. 2017). Hence, it would be informative to test the role of at least HIF-1 in this particular setting. Although a genetic approach, using siRNA or Crispr/Cas9 is perhaps technically challenging, a chemical HIF inhibitor could be used. Along these lines, to further solidify the independence of glucose of the NK cell activation, an inhibitor of glucose transporters can be used.

Could the authors speculate which fuel NK cells might use instead of glucose to fulfill the metabolic demands upon cytokine activation? 

Minor:

Among all PhD inhibitors DMOG is among the least specific. As a consequence, it is doubtful whether this experiment sheds light on the role of hypoxia in NK cell activation. Using more specific inhibitors or removing these results, is recommendable.

Author Response

Point-by-point responses to reviewer 1 comments

We thank the reviewer for the critical assessment of our manuscript and address the comments that, from our point of view, enabled us to significantly improve the manuscript.

Reviewer 1

The manuscript by Velasquez et al. investigates the early cytokine response of human NK cells during oxygen- and glucose deprivation. The major findings are:

  • Treatments with IL-15 for 6 and 16 h are both effective in priming early cytokine release in human NK cells in response to secondary stimulation with IL-12/IL-18.

  • Short-term priming induces the release of IFNγ and,additionally, CCL3, CCL4, and CCL5 from both normoxic and hypoxic NK cells in a glucose-independent manner.

  • Short-term priming of human NK cells is not associated with glycolytic switching

The manuscript is well written, the experiments thoroughly executed and the study is certainly of interest, even more so since the role of low oxygen in NK cell metabolism and function in the literature is quite controversial if not confusing. Yet, the manuscript rather adds to this confusion and does not really clarify the actual controversy.

Response:
We fully agree with the reviewer that controversy remains in the field of NK metabolism. Compared to our study, previous studies on this matter pursued broader claims and investigated NK immunometabolism in relation to more than one type of stimulus (e.g. cytokine and NK receptor stimulation), to several effector functions (cytokine response and cell killing) or signaling pathways (above all mTOR signaling). While this has delivered invaluable insights, the importance of timing of different stimuli for the usage/allocation of specific metabolic pathways and fuels for specific NK effector functions has not been addressed systematically, adding to the above mentioned confusion. With our study, we wish to go a step into this direction.

Moreover, the manuscript stays rather descriptive and does not provide a lot of mechanistic insight.  

Response:
As a reductionist approach using isolated cells and culture conditions that we manipulated in a controlled manner, our study inherently does more than observe and describe an as-is situation. We understand that the reviewer hints at the potential for gaining even more process-understanding by, e.g., engineering cellular gene expression, a particular challenge in NK cells, or by more comprehensive comparisons of cellular stimuli and their effects on several metabolic and signaling pathways as ell as effector functions. However, we feel confident that, by keeping our study focused, we provided reliable insight on the role of one particular important metabolic pathway for one critical NK effector function in response to one type of important stimulus. We hope that we could convince the reader that our in vitro replication yielded results that are very likely of physiological relevance.

The impact of the study is likely to increase by addressing the following issues:

  1. The above-mentioned confusion is largely due to different experimental setups to study the impact of hypoxia on NK cell function and metabolic changes. The authors use a set of sophisticated but complex but somewhat unique incubation scheme (e.g. different priming schemes followed by secondary activation) that renders the findings hardly comparable to previous findings on the effect of hypoxia on NK cell activation (e.g. Sarkar et al. 2013). Classical NK activation assays with IL-12/IL-18 stimulation subsequent IFNγ measurement run over 4 to 6 hours. The authors could include this experimental into their study. Although, rather simple, this layout is widely used to study NK activation.

Response:
To stress the importance of NK cell priming through IL-15, we now introduce this term already in the first paragraph of the introduction where we extend the last sentence as follows. The newly added text is underlined..

New line 31–34: “…NK cell development, survival, and homeostasis depend on interleukin 15 (IL-15) [5,6] which also promotes NK cell migration [7,8] and enhances their effector functions in response to secondary stimulation, referred to as priming [9-12].”

IL-15 priming of human NK cells for 6 h has been used by others (e.g., Kim et al., (2011) Blood 118, 5476–5486) and was adapted by us already in our previous work on the transcriptional hypoxia response in human NK cells (reference #7: Velásquez et al., 2016). In lines 43–50 of the Introduction, we introduce reports showing that mouse NK cells can reach inflamed tissues within 4–16 h. This and the reported short-term human NK cell responses to IL-15 make 6 h a reasonable time-frame. A direct comparison to longer, >16 h exposure times that induce glycolytic switching, should be the subject of future studies. Here, we focused on short-term secondary stimulation that – in agreement with the concept of priming – must succeed in time. The choice to use IL-12/IL-18 for 4 h was based on a classical scheme as stated by the reviewer. All three possible combinations of IL-15, IL-12, and IL‑18 in pairs of two are included in Figure 2 (IFNγ) production, including the combination of IL-12 and IL-18 for 4 h (Scheme 1) and 6 h (Scheme 2) as requested by the reviewer (2nd condition from the right in each scheme). All three combinations led to equally suboptimal IFNγ production as already stated on new lines 216–217: “…The three combinations of two cytokines each appeared to, on average, expand the population of IFNγ+ cells from below 5% to around 20%. ...”

         Sarkar et al. used in their 2013-study both hypoxic and anoxic culture conditions but combined cytokine stimulation only with anoxia, not with hypoxia. A major difference between anoxia and hypoxia is that hypoxia provokes the cellular production of superoxide anions by the respiratory chain. HIF-1α mediated induction of pyruvate dehydrogenase kinase isozyme 1 (PDK1), which is also observable at the transcriptional level in primed hypoxic human NK cells (Velásquez et al., 2016), is well known to protect cells from oxygen radical production by restricting pyruvate dependent TCA cycler activity. Hence, anoxia is likely not comparable to hypoxia in its effect on NK cells.

  1. Along these lines, in addition to cytokine release, the actual killing of target cells is a hallmark feature of NK cells. The authors should therefore perform killing assays under their particular incubation scheme. Even more so since the impact of hypoxia on NK cell function/cytotoxicity seems to be most obvious in an actual killing assay upon exposure to target cells (Sarkar et al. 2013). Likewise, in addition to cytokine activation, NK cells are activated or inhibited in a ligand-dependent manner. Ideally, this could be addressed.

Response:
We concur with the reviewer that, in addition to cytokine release, the actual killing of target cells is a hallmark feature of NK cells. But there is a timeline to the events from NK cell migration to target sites to communication with endothelial cells and other immune cells, killing of stressed cells, and support of transition to an adaptive immune response and of resolution of inflammation (as discussed in new lines 528-531 and in the Supplemental Discussion). In this timeline, our study focuses on events and functions of NK cells prior to target cell contact, namely, early IFNγ and CC chemokine production that drive not only the adaptive but also already the innate immune response. In our view, the metabolic requirements of cell killing are a subsequent story. We believe that we must formulate our hypotheses and design our experiments in a way that dissects the processes that govern NK cell function during inflammation and the pertaining questions before synthesizing them again. It is also technically non-trivial to obtain conclusive information from the study of cell killing under adverse environmental conditions that at the same time alter target cell susceptibility to NK killing. Above all, it is well known that hypoxia induced autophagy can protect tumor cells from perforin-granzyme mediated cell killing. Therefore, we feel that additional experiments on cell killing would by far exceed the scope of our study and rather justify a new, dedicated investigation.

  1. The adaptive response to low oxygen depends on the HIF transcription factor family, with HIF1 and HIF2 as the most prominent members. This pathway also seems to play a role in NK cell activation in normoxia and hypoxia (Krzywinska, et al. 2017). Hence, it would be informative to test the role of at least HIF-1 in this particular setting.

Response:

Curiously, the anti-tumor effect of the NK-specific HIF1A knockout in mice reported by Krzywinska et al. (2017) stems from a reduced number of  NK cells that reach the tumor and thus a reduced amount of NK cell produced angiostatic soluble VEGFR-1 resulting in a less productive angiogenesis and thus reduced tumor growth. Contrarily, Ni et al. from the Cerwenka group characterize NK cell HIF-1α as an immune checkpoint protein (see conference abstract at https://cancerres.aacrjournals.org/content/78/13_Supplement/4743). We believe that a better understanding of the animal systems used is required to bring these findings into accord.

Yet, we agree with the reviewer that direct detection of the HIF-1α response and understanding its interaction with immune signaling in NK cells is important. Using a systems immunology approach, we recently identified an IL-15/STAT3/HIF1A signaling axis, and potentially a IL-15/NF-κB/HIF1A axis, in human NK cells in addition to the well known IL-15/mTOR/HIF1A axis (reference #50: Coulibaly et al., 2019).  While in that study we used chemical hypoxia to stabilize HIF-1α, we are currently investigating the functional and metabolic importance of this signaling axis using actually low oxygen (see conference abstract #50 for preliminary results at https://files.lih.lu/events/2019/NK2019_abstractbook.pdf). As outlined above, we require a better understanding of the signaling interaction between cytokine and hypoxia before coming up with hypotheses on the role of HIF1A in immunometabolism which is still obscure.

  1. Although a genetic approach, using siRNA or Crispr/Cas9 is perhaps technically challenging, a chemical HIF inhibitor could be used.

Response:
Given the technical challenge to genetically manipulate primary NK cells, we are currently using different signaling pathway inhibitors that in part have also demonstrated anti-tumor activity (namely, STAT3 inhibitors). Unfortunately, to our knowledge the available inhibitors of HIF-1α transcriptional activity have indirect mechanisms of action, e.g., chetomin which prevents interaction of HIF-1α with its coactivator p300 that also interacts with many other transcription factors. In our hands, chetomin was highly toxic to human NK cells (reference #7: Velásquez et al., 2016). Therefore, we expect available HIF-1α inhibitors to show low specificity.

  1. Along these lines, to further solidify the independence of glucose of the NK cell activation, an inhibitor of glucose transporters can be used.

Response:

Chemical inhibition of glucose transporters would require prior testing for non-toxicity and effectiveness. But in the light of the effective rapidity in the decay of glycolytic flux upon glucose deprivation as demonstrate in Figure 6, we do not believe that chemically blocking of cellular glucose uptake will yield additional insight.

  1. Could the authors speculate which fuel NK cells might use instead of glucose to fulfill the metabolic demands upon cytokine activation? 

Response:
Dendritic cells have been demonstrated to draw on glycogen stores upon activation (Thwe et al., 2017, Cell Metabolism 26, 558–567) and granulocytes rely on glucose stored in the endoplasmic reticulum (Jun et al., Blood. 2010;116(15):2783-2792). Both these processes yield energy, independent of environmental glucose availability, but eventually still require glycolysis. As our flux analyses (especially Figure 6) strongly argue against this possibility in human NK cells, we already speculated at the end of our manuscript, that the creatine kinase energy system may play a role in meeting metabolic demands upon cytokine activation.

New line  658–660: “…alternative ways of energy storage may function in NK cells, such as the creatine kinase energy system, which supports production of IL-2 and IFNγ in T lymphocytes [65].”

Minor:

  1. Among all PhD inhibitors DMOG is among the least specific. As a consequence, it is doubtful whether this experiment sheds light on the role of hypoxia in NK cell activation. Using more specific inhibitors or removing these results, is recommendable.

Response:
Although DMOG is highly effective in activating HIF-1α and glycolysis, its detrimental effect on cytokine production makes it indeed unsuitable for the study of NK cell activation. However, DMOG still has potential as an anti-inflammatory drug (as discussed in new line 624–625), and does not necessarily reduce in vitro lymphocyte cytokine responses. DMOG treated T lymphocytes, for instance, release more IL-17A (Bollinger et al., J. Leukoc. Biol. 96: 305–312; 2014). Therefore, understanding the effects of this compound is still of interest, and we believe that it is justified to keep our data on the cytokine response in human NK cells in the presence of DMOG as a supplementary result in the manuscript (now Supplemental Figure S6).

Reviewer 2 Report

The authors describe that the early release of IFNγ and chemokines by IL-15-primed-IL12/IL18 stimulated human NK cells is not a metabolically controlled process. In particular, the authors analyze the role of glucose and hypoxic conditions demonstrating that the early NK activation is independent on these two factors. The work is well executed, but some controls are missing:

  • Please, add flow cytometry strategy for Figure 2, Figure 4 and Figure 8.
  • In all the seahorse experiments a "not treated" condition is missing (NK cells w/o IL-15, IL-12 or IL-18 stimulation). The authors should show both ECAR and OCR of this group as control.
  • A real control for hypoxia is missing. The author should provide an immunoblot of HIF-1a protein and qPCR of HIF-1a targets (Egln3)
  • Chemical inhibitors have off-targets and the doses used in Figure 3 probably are too low. In order to better analyze the contribute of glycolysis and glutaminolysis, I suggest to use glucose and glutamine free media.
  • In Figure 6, the authors should show also OCR data and not treated conditions.
  • Figure 7 and Figure 8 are key experiments. The authors should show hypoxic validation in Figure 8 (HIF1a blot or an equivalent control). Low oxygen fails to induce an increase of glycolysis in human NK cells, making difficult the interpretation of these data in the role of glycolysis in NK activation. I suggest to use oligomycin, FCCP or Antimycin A to block mitochondrial respiration and force glycolysis. In this manner the authors could validate their conclusion about the glycolytic independency in early human NK activation.   

Author Response

Point-by-point responses to reviewer 2 comments

We thank the reviewer for the critical assessment of our manuscript and address the comments that, from our point of view, enabled us to significantly improve the manuscript.

Reviewer 2

The authors describe that the early release of IFNγ and chemokines by IL-15-primed-IL12/IL18 stimulated human NK cells is not a metabolically controlled process. In particular, the authors analyze the role of glucose and hypoxic conditions demonstrating that the early NK activation is independent on these two factors. The work is well executed, but some controls are missing:

  1. Please, add flow cytometry strategy for Figure 2, Figure 4 and Figure 8.

Response:
This is an important request in response to which we modified the text at the end of the Methods section on “2.4. Flow Cytometry Analysis” and included an according graphical illustration (Supplemental Figure S2).

New lines 186–189 “…We gated on singlets in the forward scatter area versus height plot followed by gating on lymphocytes in the sideward scatter area versus forward scatter area plot. We applied sequential gates on biparametric dot plots to calculate marker percentages and median fluorescence intensity (MeFI) values in the CD56+ population (Supplemental Figure S2).

  1. In all the seahorse experiments a "not treated" condition is missing (NK cells w/o IL-15, IL-12 or IL-18 stimulation). The authors should show both ECAR and OCR of this group as control.

Response:

Resting NK cells are well known to display extremely low levels of glycolysis and mitochondrial respiration as introduced on lines 62–63 (Introduction). Yet, we agree with the reviewer that it is of interest to include here a demonstration of the metabolic effect of IL-15 priming alone on human NK cells. Because the capacity of the Agilant Seahorse XFp analyzer instrument available for this study allows parallel measurements in only 6 wells at a time, technically limiting our options for multiple comparisons, we choose to compare 6 h of IL-15 priming versus no treatment in a separate experiment. As already predicted from our previous results (reference #7: Velásquez et al., 2016, see line 79 and 80) the increase in glycolysis through priming was indeed moderate. Hence, we include this measurement in the Supplementary Material. The new Supplemental Figure S3 shows traces for ECAR and OCR (panel A) as well as glycolytic parameters (panel B) for untreated and primed human NK cells from 3 donors. We now refer to the new Supplemental Figure S3 in the main text in section “3.4. Moderate Early Increase in NK Cell Glycolysis through Hypoxia and Inflammatory Cytokines” and in the Discussion as follows

New line 335–336 “…Priming of resting NK cells for 6 h with IL-15 led to very low but still noticeable elevations of basal glycolysis and also respiration (Figure S3)…”

New line 613–615 “… As priming itself, hypoxic versus normoxic human NK cells, however, showed only moderately increased glycolysis following priming (Supplemental Figure S3 and Figure 5A),…”

  1. A real control for hypoxia is missing. The author should provide an immunoblot of HIF-1a protein and qPCR of HIF-1a targets (Egln3)

Response:

Whereas the positive effect of hypoxia on glycolysis in Figure 5A can readily be interpreted as cellular adaption to hypoxia, we agree with the reviewer that an independent hypoxia control is desirable for the cytokine response which is at the center of this work. From the experiments reported in the submitted work, there are however no samples of cellular protein or RNA available for further transcriptional or immunoblot analyses. Instead, we propose to refer to the hypoxia induced upregulation of GLUT1, a well-known component of cellular adaption to hypoxia and shown in Figure 4B and 4C, as an internal control for the test of the effect of hypoxia on the cytokine response shown in Figure 8. The GLUT1 analyses in Figure 4 and the cytokine analyses in Figure 8 were recorded in same samples from the same experiment. We made the following according adjustments to the text in section “3.6. Hypoxia Hardly Affects Early Cytokine Induced IFNγ Production and Chemokine Release by NK Cells”. The newly added text is underlined:

New line 425–429 “…we assessed the influence of hypoxic versus normoxic culture of human NK cells on the IL-12/IL-18 induced production of IFNγ and, additionally, the release of CCL3/4/5 after 6 h of IL-15. The experiment was conducted using the same cultures as for the assessment of GLUT1 surface expression (Figure 4B,C) the upregulation of which is a sign of functional cellular adaption to hypoxia. However, hypoxia affected neither…”.

In this context, we would like to point out that the HIF-1α dependent hypoxia response of IL‑15 primed human NK cells is very robust on the transcriptional level, i.e., elevated expression of target genes and pathways (see line 80 and 81, reference #7: Velásquez et al., 2016). Moreover, we are currently completing another project on the regulation of HIF-1α through IL-15 signaling in hypoxic human NK cells (preliminary data presented as a poster at the NK2019 conference, abstract #50 available at https://files.lih.lu/events/2019/NK2019_abstractbook.pdf.

  1. Chemical inhibitors have off-targets and the doses used in Figure 3 probably are too low. In order to better analyze the contribute of glycolysis and glutaminolysis, I suggest to use glucose and glutamine free media.

Response:

Several papers, especially on transformed cells, indeed report the use of higher concentrations of the metabolic pathway inhibitors UK5099, BPTES, and etomoxir than given in the legend to Figure 3 and recommended in the Agilent Seahorse XF Mito Fuel Flex Test kit. On their web site (https://community.agilent.com/docs/DOC-9195-agilent-seahorse-mito-fuel-flex-report-generator-faqs), Agilent states that “…the test uses all three compounds at concentrations well above their EC50 values for inhibition in mammalian cells. These values have been validated in a variety of cell lines and primary isolates. …” We found a report on IC50 values for the inhibition of palmitic acid oxidation for etomoxir in various cell systems (Wensaas et al., J. Lipid Res. 2007. 48: 961–967, see Figure 6) that supports this notion for the fatty acid oxidation pathway inhibitor.

Although overall robust, human NK cells in our hands tolerate certain inhibitors only at concentrations at least one order of magnitude below what is accepted by transformed cells, e.g., DMOG (reference #50: Coulibaly et al., 2018). The HIF-1α inhibitor chetomin was not tolerated at all at presumably active concentrations (reference #7. Velásquez et al., 2016). From this perspective, we expect the application of higher concentrations of metabolic pathway inhibitors in human NK cells to require extensive testing for cytotoxicity.

We thank the reviewer for the suggestion to use of fuel-free media to also assess specific fuel contributions to respiration. We now include the OCR traces from our glucose withdrawal experiment shown in Figure 6 as Supplemental Figure S5. And we adjusted the text in the according section “3.5. Early Cytokine Induced IFNγ and Chemokine Responses in NK Cells do not Depend on Glucose”, because the data needs to be correlated to the corresponding ECAR traces shown in Figure 6 within this section, as follows.

New lines 389–392 “Notably, the OCR traces corresponding to D1–D3 from the experiment shown in Figure 6 show similar and stable respiratory activities over time both with and without glucose (Supplemental Figure S5), which agrees with no important role of pyruvate as a mitochondrial fuel (cf. Figure 3).”

The focus of the submitted study was to explore the short-term importance of glucose because glycolysis was shown to become essential for NK effector functions in the longer run, and because glucose is expected to gain in importance under hypoxia. Before assessing the functional consequences of low glucose and low oxygen, we therefore tested for correlations between IFNγ inducing conditions, i.e., IL-15 priming and secondary IL-12/IL-18 stimulation, with a number of selected cellular readouts presumably indicative of changes in the importance of glycolysis. Besides mitochondrial fuel selection and glycolytic flux, these also included glycolytic protein expression (see Figure 4). We believe that an equally thorough assessment of the importance of glutaminolysis and fatty acid oxidation for NK effector functions must also look beyond fuel selection and consider changes in the quantities of metabolic intermediates, alternative to flux measurements, e.g., by isotope labeling and mass spectrometry (MS), as well as detection of metabolic proteins (enzymes, transporters etc.) as shown in Figure 4 for glycolytic proteins. An analytical assessment of the importance of glutamine and fatty acids, in a manner equivalent to glucose, thus exceeds the scope of our current study. Note that even when flux through the glycolytic pathway is high, absolute concentrations of glycolytic intermediates is low and hardly accessible to detection in human cells by MS. Hence, in response to the concern by the reviewer, we included the following text in the Discussion.

New lines 588–591: “…Before, it needs to be mentioned that, as a limitation, our study does not present a conclusive assessment of the importance of glutaminolysis and fatty acid oxidation for human NK effector functions. This will require determining changes in the quantities of metabolic intermediates, e.g., by mass spectrometry, as well as of the involved metabolic proteins. …”

  1. In Figure 6, the authors should show also OCR data and not treated conditions.

Response:

We introduced an according change already in response to comment #11 (see Supplemental Figure S5 and text in lines 388–390). Basal OCR values for untreated versus primed NK cells are included in Supplemental Figure S3.

  1. Figure 7 and Figure 8 are key experiments. The authors should show hypoxic validation in Figure 8 (HIF1a blot or an equivalent control).

Response:

We introduced hypoxia induced GLUT1 expression (Figure 4B,C) as an equivalent control already in response to comment #3, which anticipates comment #6.

  1. Low oxygen fails to induce an increase of glycolysis in human NK cells, making difficult the interpretation of these data in the role of glycolysis in NK activation. I suggest to use oligomycin, FCCP or Antimycin A to block mitochondrial respiration and force glycolysis. In this manner the authors could validate their conclusion about the glycolytic independency in early human NK activation.   

Response:

The difference between the increase in glycolytic flux in IL-15 primed human NK cells through oligomycin, i.e., the glycolytic reserve, is somewhat larger than through low oxygen but still very small (Figure 5A and text in lines new 337–338: “...Figure 5A shows a slight increase in glycolytic rate and capacity through hypoxic versus normoxic IL-15 priming for 6 h. …”). But notably, IL-15 priming abolishes glycolytic reserve in IL-12/IL-18 stimulated cells, i.e., in IFNγ inducing conditions (Figure 5B, text in new lines 341–343: ”… IL-15, however, also appeared to prevent further enhancement of capacity through secondary IL-12/IL-18 stimulation under normoxia…), in which glycolysis can thus apparently not be further forced. This, however, agrees with the repeatedly cited observation by Keppel et al. (2015), that oligomycin did not affect IFNγ production in mouse NK cells treated with IL-12/IL-18 for 6 h (see reference #49, specifically, in new lines 80 and 606). We modified the following sentence in the Discussion accordingly (the additional new text is underlined).

New line 621-624 “…This data and the lack of additional glycolytic reserve through secondary stimulation in primed compared to non-primed human NK cells also conform to unchanged IFNγ positivity in mouse NK cells treated for 6 h with IL-12/IL-18 in the presence of oligomycin [49]. …”

Round 2

Reviewer 1 Report

The authors ave addressed some of the concerns by discussing their results more extensively in their rebuttal letter (not in the manuscript, though) but seem to be reluctant to perform additional experiments. Given the overall controversy in the field, I still believe that additional experiments are helpful to improve the quality of the manuscript.
